# Throughput screening of *Bacillus subtilis* strains that abundantly secrete surfactin *in vitro* identifies effective probiotic candidates

**Dongho Lee**[1], **Taehoon Oh**[2], **Bongseok Kang**[3], **Jong Seok Ahn**[2]*, **Yangrae Cho**[3]*

**1** Molpaxbio, Yuseong-gu, Daejeon, Republic of Korea, **2** Chemical Biology Research Center, Korea Research Institute of Bioscience and Biotechnology, Ochang-eup, Cheongwon-gu, Cheongju-si, Chungcheongbuk-do, Republic of Korea, **3** ProxEnrem, Osong-eup, Chungju-si, Republic of Korea

* yangchorae@gmail.com (YC); jsahn@kribb.re.kr (JSA)

**Data Availability Statement:** Their genome sequences were deposited in GenBank. Accession numbers for ps4060 and ps4100 were CP081458 and CP076445, respectively. The strains were also

## Abstract

Since the prohibition of antibiotics as animal growth promoters, demand for effective probiotic strains has steadily increased. The goal is to maintain productivity and mitigate environmental concerns in the livestock industry. There are many probiotic animal-diet supplements available, over 2,000 products in the Republic of Korea alone, with little explanation about the desirable properties of each probiotic strain. The purpose of this study was to describe the underlying logic and methods used to select two novel strains of probiotic candidates. To economically screen these candidates, the abundance of surfactin secreted was used as an *in vitro* marker. We used a modified oil-misting method to screen ~2,000 spore-forming bacteria for novel strains of *Bacillus subtilis*. Of these, 18 strains were initially selected based on the semiquantitative criterion that they secreted more surfactin than *B. subtilis* ATCC21322 on Luria-Berani (LB) agar plates. The whole genome sequence was determined for two of the 18 strains to verify their identity. A phylogeny of 1,162 orthologous genes, genome contents, and genome organization confirmed them as novel strains. The surfactin profiles produced by these two strains consisted of at least four isoforms similar to standard surfactin and enhanced cellulase activities up to 50%. Four fractionated individual isoforms of surfactin suppressed inflammation induced by lipopolysaccharides. The half-maximal inhibitory concentration ($IC_{50}$) was about 20 μM for each isoform. Both selected strains were susceptible to seven important antibiotics. Our results implied that an abundant secretion of surfactin was a useful biomarker *in vitro* and could be utilized for mining probiotic candidates through high-throughput screening of environmental samples.

## Introduction

Multidrug-resistant bacterial strains and environmental pollution that originates from intensive animal farming could become serious social and medical problems. Such concerns led the industry from a reduction in the use of antibiotics to their total ban as animal growth promoters, including a massive reduction as therapeutics [1]. Beneficial microorganisms known as

deposited in the Korea Microbe Collection Center with accession numbers KACC81161BP and KACC81159BP respectively for ps4060 and ps4100.

**Funding:** This research was partly supported by two anonymous persons unknown to YC, the R&D fund of the Cheongbuk-do and Cheongju-si local government to YC, and the Technology Development Program (S3089057) of the Ministry of SMEs and Startups (MSS, Korea) to YC. It was also partly supported by the National Research Council of Science & Technology (NST) granted by Research and Development for SMEs project to JSA. There was no additional external funding received for this study.

**Competing interests:** YC owns the startup company, ProxEnrem Inc., where he identified the Bacillus strains as candidates for animal diet supplements. We do not anticipate an immediate financial benefit from this publication, but it may be easier to acquire a permit to register the strains for animal diet supplements. This work was "partly" supported by two anonymous persons who were willing to assist technology development for a reduction in environmental pollution and somewhat compensate for the inevitable production of CO2 during industrial development in the Republic of Korea. They will not have a direct financial benefit from this publication. Neither did they influence the study design or data interpretation. The strains Bacillus subtilis ps4060 and ps4100 were protected by a Korean patent. Private financial assistance or patent protection does not alter our adherence to PLOS ONE policies on sharing data and materials.

probiotics have emerged as growth promoters and viable alternatives to antibiotics and harmful chemicals. Spore-forming bacteria are potentially promising probiotics [2]. Unlike vegetative cells that are usually killed by gastric and bile fluids, bacterial spores survive the harsh conditions of the gastrointestinal tract and reach the large intestine of farm animals and humans [3, 4]. They germinate in the digestive tract and newly germinated vegetative cells persist without active proliferation [4–6]. Vegetative cells would remain metabolically active, secreting bacteriocins important for competing with pathogenic bacteria in the digestive tract [7].

Important properties of probiotic strains for livestock are directly associated with enhancing productivity and animal welfare, and possibly the reduction of noxious gas emission resulting from a decreased production of methane, ammonia, hydrogen sulfide, and carbon dioxide [8]. Several bacterial genera have been developed as probiotics, including *Lactobacillus*, *Bifidobacterium*, and *Bacillus*. Interest in *Bacillus* spp. as new probiotics has been steadily increasing, partly because they form spores. Whole-genome sequences for 11,613 *Bacillus* strains were listed in NCBI on May 2, 2022. The list included 1,016 strains of *B. subtilis*, 427 strains of *B. licheniformis*, 263 strains of *B. amyloliquefaeciens*, and 855 strains of *B. velezensis*. Some of these strains have been developed as probiotics. Nonetheless, the need for probiotics is growing, and screening novel strains with high efficacy is still necessary.

Functional testing *in vivo* to evaluate their probiotic effect on the animal host is a long, expensive effort compared to *in vitro* testing. It is therefore beneficial to select a small number of promising candidates by massive *in vitro* screening before performing *in vivo* tests. Characteristics associated with *in vivo* function, and that can be evaluated by *in vitro* tests, however, are poorly defined. We decided for several reasons to test whether the abundant secretion of surfactants *in vitro* was a potential biomarker for *in vivo* functionality. First, biosurfactants secreted by several *Bacillus* species have antimicrobial properties comparable to conventional chemical drugs [7]. They include lipopeptides like surfactin, fengisin, and iturin. We hypothesized that the abundant secretion of antibacterial surfactants was a desirable feature for suppressing infectious microbes and preventing diseases in animals. It may have an effect similar to antibiotics. Although secretion of any antimicrobial peptide would prevent the colonization of the intestine by pathogenic bacteria, we preferred to screen surfactant-secreting strains because surfactin also has anti-viral, anti-fungal, and anti-mycoplasma activities [9].

For high-throughput screening, we adopted the oil-misting method of Burch and colleagues [10]. This method does not require a liquid culture step and the results are straightforward and fast to analyze. In this study, we discovered two novel *B. subtilis* strains that abundantly secrete biosurfactants *in vitro*. We then examined the properties of surfactin that pertain to probiotic functions, and discuss possible reasons for considering them as potent probiotic candidates.

## Materials and methods

### Screening spore-forming and surfactant-producing bacteria

To discover surfactant-producing bacteria among heat-tolerant microorganisms, a large number of microbial colonies were screened by a modified airbrush method [10]. First, 0.5 g of a dried environmental sample, such as soil or animal feces, was ground by mortar and pestle. The powder was put in a tube suspended in 50 mL of distilled water preheated to 80°C, followed by incubation in an 80°C water bath for 5 min. The tube was then centrifuged at 3,000 g for 10 min to precipitate the heavier substances. The slightly opaque layer of supernatant formed was spread on LB plates and incubated at 37°C overnight. Next, well-separated bacterial colonies from the LB plates were spotted onto fresh LB agar plates using sterile yellow pipette tips and incubated overnight. *Bacillus subtilis* strain ATCC21322 was also inoculated

onto each plate as a positive control. An airbrush, Beetle Bug (Yamato comp., Seoul, ROK), was used to apply a fine mist of canola oil onto the plates at an air pressure of 2 bars. The distance between the air brush and plate was set to deposit a fine mist of oil droplets, making biosurfactant halos visible to the eye without special equipment. The size and transparency of the halo correlated with the amount of secreted surfactant [10]. Halo quality was visually assessed by clarity, transparency, and evenness of the oil layer in the clear zone, using three grades: 1) opaque with numerous small oil drops, 2) medium clear with a small number of oil drops, and 3) clear with no visible oil drops. Halo size was measured as the diameter of the clear zone from the leading edge, through the center to the other-side edge.

## Genome sequencing, assembly, initial annotation

Genome sequences of two *B. subtilis* strains, ps4060 and ps4100, were determined and assembled at DNA-Link (DNA-Link, Seoul, ROK) using the PacBio platform (Pacific BioScience, Menlo Park, CA) following manufacturer's instruction. We used 5 μg of each sample to construct a SMRTbell library with the SMRTbell™ Template Prep Kit 1.0 (PN 100-259-100). A sequencing primer was annealed to the SMRTbell template over a 20-kb genomic DNA fragment to construct a sequencing library [11]. The library was sequenced using 1 SMRT cell with MagBead OneCellPerWell v1 Protocol for 240 minutes using the PacBio RS II (Pacific Biosciences) sequencing platform. Over 64,000 long reads and 570 million base pairs were identified for each strain. After filtering subreads, the average sequence coverage of the genome was over 110 times. *De novo* assembly was conducted using the hierarchical genome assembly process (HGAP, Version 2.3) workflow. This process included consensus polishing with Quiver [12]. Since bacterial genomes and plasmids are typically circular, we checked the form of each contig using MUMmer 3.5 [13] and trimmed one of the self-similar ends for manual genome closure. The genome sequences were deposited in GenBank with accession numbers CP081458 and CP076445 for ps4060 and ps4100, respectively. The strains were also deposited in the Korea Microbe Collection Center, Korean Agricultural Culture Collection (KACC, http://genebank.rda.go.kr/), National Institute of Agricultural Science International Depository Authority (https://www.acm-mrc.asia/M/kctc.html), with accession numbers KACC81161BP and KACC81159BP respectively for ps4060 and ps4100.

## Gene phylogenies and comparison of bacterial genomes

The genomes of ps4100 and ps4060 each contained ten rRNA operons. The ten 16S rRNA sequences were mined from each genome and examined: only four of them had unique sequences. Other *Bacillus* species also had multiple copies of rRNA operons in their genomes, from which only unique sequences were subsequently used to construct gene-phylogenies. Accordingly, one copy among identical sequences was used for phylogenetic tree construction. Ribosomal RNA generally reflects organismal phylogeny; however, this gene is also transferable among closely related species [14, 15]. In addition, there were insufficient phylogenetic signals for subspecies classification. To minimize rare but possible gene transfer effects on organismal phylogeny and to acquire subspecies-level resolution, additional phylogenies were constructed with single-copy genes shared among *B. subtilis* strains. Initially, 430 whole-genome sequences were downloaded from NCBI. From these we selected 176 taxa, each with a genome of less than four contigs. The genomes of ps4060 and ps4100 were included in subsequent analyses. Each genome was annotated by the NCBI Prokaryotic genome annotation pipeline (Version 6.1 March 2022). Candidates with common genes were clustered using options of: 70% sequence identity, a 0.9 length difference cutoff, 0.9 alignment coverage, and a 90% length difference imbedded in CD-HIT [16]. After selecting 1,162 clusters, individual

gene sequences were concatenated using Python script, followed by multiple sequence alignment using UGENE or MAFFT v4.763 [17, 18]. Gaps were removed before phylogenetic analysis using the maximum likelihood method in MEGA [19]. Further, a phylogeny with 13 taxa was constructed with those representing major groups in the *B. subtilis* phylogeny and Bootstrap analysis was also performed. For visual comparisons of whole genome structure, global alignment of bacterial genomes between most-closely related strains was performed with Mauve [20]. Mauve aligns multiple genome sequences and is especially instrumental for showing a large-scale inversion and genome rearrangement. Options used are 'align with progressiveMuave, default seed weight, determine LCBs, default mean LCB weight, full alignment, iterative refinement, and sum-of-pairs LCB scoring'.

## Production and quantification of surfactin

For surfactin production, we inoculated a 250-mL flask containing 50 mL of surfactin production medium with a loopful of the bacteria. The medium contained 40 g/L glucose, 5 g/L yeast extract, 10 g/L peptone, 3.27 g/L $K_2HPO_4 \cdot 3H_2O$, 1.5 g/L NaCl, 0.5 g/L $Na_2CO_3$, 1 g/L $MgSO_4$, and 0.91 g/L $FeSO_4 \cdot 7H_2O$. This composition was determined based on our fermentation experience and three publications [21–23]. The flask was agitated by shaking at 150 rpm and 30˚C for 70 h. The culture was transferred to 50-mL centrifuge bottles, centrifuged at 10,000 g, and a cell-free broth recovered by removing the cells and cell debris. The broth was acidified to pH 2–4 by adding 1 M HCl and incubated at 4˚C overnight. A second centrifugation at 13,000 g for 20 min was used to harvest the brown precipitants. The precipitants were stored at 4˚C or -20˚C until use, or directly dissolved in 95% methanol for downstream quantification. Surfactin concentration was determined by reverse phase HPLC equipped with a C18 column (column diameter 5 μm) at 30˚C [24]. The samples were filtered through a 0.45-μm membrane (Corning, Berlin, Germany) before analysis. A mixture of trifluoroacetic acid (20% v/v) and acetonitrile was used as the mobile phase. The flow rate was 1.0 mL/min. Quantitative analysis of each sample was conducted at least twice under identical conditions. Surfactin purchased from Sigma-Aldrich (St. Louis, MO, USA) was used as the standard.

## Effect of surfactin on cellulase activity

Surfactin was completely dried and dissolved in methanol (10 mg/mL). The resulting solution was subsequently diluted with deionized water or an appropriate buffer as described for each experiment. Surfactin solution was added to the reaction mixture and cellulase activity measured [25]. The cellulose substrate was dissolved in a large volume of 50 mM acetate buffer (pH 5.0). An addition of deionized water without buffer, or a small amount of 1 M Tris-HCl buffer (pH 8.0) to make a 3 mM concentration, did not affect the pH of the reaction mixture. Spectrophotometric Stop Rate Determination was performed following the protocol of Enzymatic Assay of Cellulase (Sigma-Aldrich, St. Louis, MO, USA). The cellulase solution, 1 unit/mL in cold deionized water, was prepared immediately before the enzymatic assays. Glucose generated by the enzyme reaction was calculated by the increase in ΔΔA340 nm; (A340 nm Test final—A340 nm test initial)—(A340 nm blank final—A340 nm blank initial). We used ΔΔ to compensate for the possible irregularity of background absorbance in each well. Relative amounts of glucose were calculated by (ΔΔA340 nm of reaction with surfactin—ΔΔA340 nm of reaction without surfactin) / ΔΔA340 nm of reaction without surfactin *100.

## Analysis of nitric oxide production in macrophage cells

We fractionated four major isoforms of surfactin a, b, c, and d (S1 Fig in S1 File) from the total surfactin dissolved in 99.99% methanol by a preparative HPLC system (Agilent Technology,

Santa Clara, CA, USA). The fractionated isoforms were completely dried in a SpeedVac vacuum drier (Eppendorf, Hamburg, Germany), and then dissolved in dimethyl sulfoxide (DMSO), which was effective regardless of pH [26]. We used DMSO alone in a negative control experiment. We tested the effect of each isoform on nitric oxide (NO) production as an indicator of inflammatory responses in mouse macrophage RAW 264.7 cells [27, 28]. The RAW 264.7 cells were cultured at $5 \times 10^4$ cells/well in a 96-well microplate in 200 μL of Dulbecco's Modified Eagles Medium (DMEM). The medium contained 10% fetal bovine serum (FBS), 100 units/mL penicillin, and 100 μg/mL streptomycin. To induce an inflammatory response, we added 0.5 μg/mL lipopolysaccharide (LPS) to each well pretreated for 30 min with different amounts of a surfactin isoform. After the cells were cultured at 37°C for 24 h, they were centrifuged to separate supernatant and precipitant. The supernatant from the cell culture was recovered after centrifugation and the concentration of NO measured by the NO Plus Detection Kit (iNtRON Biotechnology, Seongnam, Republic of Korea) using the Griess reagent system following the manufacturer's protocol [29]. Absorbance was measured at 540 nm with a Spectra Max 190 microplate reader, (Molecular Devices, San Jose, USA) by converting the NO concentration in the supernatant using a standard curve for each concentration of sodium nitrite. The half-maximal inhibitor concentration ($IC_{50}$) was obtained using GraphPad Prism (GraphPad Software, 8.4.3, San Diego, CA, USA).

## Measurement of iNOS protein

The RAW 264.7 cells were treated with 0.5 μg/mL LPS to induce an inflammatory response. Surfactin isoforms A, B, C, and D were individually added to the cells at 10 and 25 μM concentrations, followed by 24 h at 37°C in the presence of 5% $CO_2$. We used a Western blot to examine iNOS expression after collecting the cells from each treatment and washing them twice with PBS. Cell lysis buffer (10 mM Tris-HCl pH 7.4, 150 mM NaCl, 1 mM EDTA, 1% Triton X, 1 mM PMSF, 1 μg/mL aprotinin, 1 μg/mL leupeptin, 1 mM DTT) was added and the cells disintegrated over 30 min, followed by centrifugation at 4°C and 13,500 rpm for 20 min. Soluble proteins were separated on an 8% SDS-PAGE gel, and then transferred to a nitrocellulose (NC) membrane (Whatman, Maidstone, UK) by electroblotting for 3 h at 250 mA. The membrane was blocked for 1 h at room temperature in TBS-T buffer (20 mM TRis-HCl pH 7.4, 150 mM NaCl, 0.05% Tween-20) containing 5% nonfat dry milk and washed with the buffer. The membrane was incubated with the primary antibody (1:1,000 dilution; ABcam, Cambridge, UK, ab15323) for iNOS at 4°C overnight, washed five times, and then incubated for another 2 h with HRP-conjugated anti-rabbit IgG as a secondary antibody. The internal control was β-actin antibody (1:1,000 dilution; Santa Cruz, CA, USA, sc-47778). After washing the membrane five times, the protein signals were visualized with SuperSignal™ West Femto Maximum Sensitivity Substrate (Thermo Fisher, Rochford, USA) and the signal intensity measured by densitometry using Image J Software (Version 1.52a).

## Antibiotics resistance tests

An Epsilometer test (E-test) was performed for quantitative minimal inhibition concentration (MIC) determination following manufacturer's protocol (bioMérieux, Stockholm, Sweden). Briefly, freeze-dried spores were diluted to $1 \times 10^8$ cfu/mL in distilled water. A sterile cotton swab was dipped into the inoculum, excess liquid removed, and then streaked over the entire surface of an agar plate. The plate was left for about 5 min to allow extra moisture to be absorbed by the agar. An E-test strip was then placed on the agar surface using forceps and the covered plates incubated at 37°C for 13 to 15 h. After incubation, an inhibition zone of bacterial growth formed as an ellipse that intersected the MIC reading scale on the μg/mL unit [30].

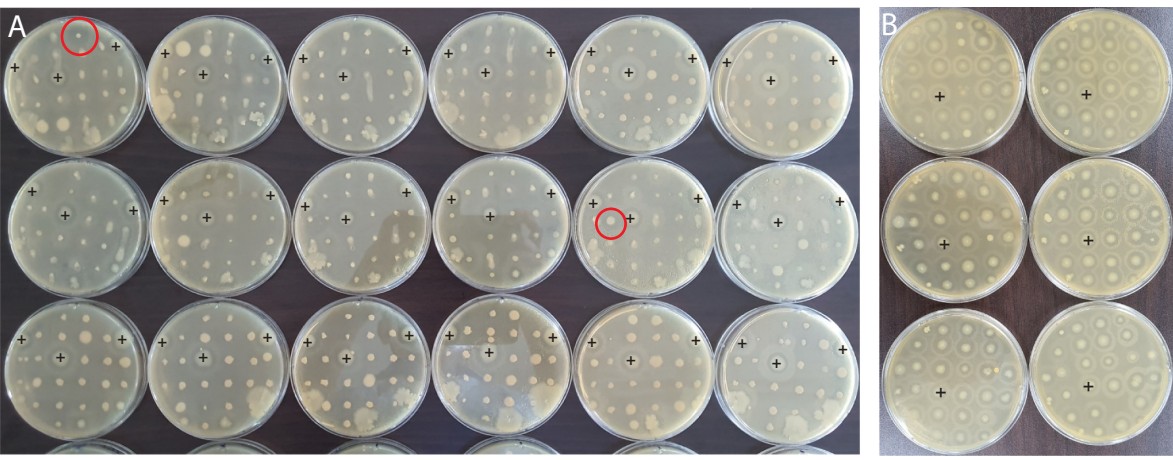

**Fig 1. Initial screening of surfactin-producing bacteria.** A. First round of screening: Most bacterial colonies did not produce a clear zone after oil-misting, with a few exceptions and the positive controls (+). Colored circles indicate two colonies with larger, clearer halos than the control. B. Second round of screening: Most colonies produced larger or clearer halos than the positive controls.

## Results

### Initial screening of strains with abundant surfactant secretion

A large number of heat-tolerant bacteria from a personal collection of insect feces and soils from remote mountains were screened to discover novel strains of surfactant-producing *Bacillus* species. A fine mist of oil droplets spread evenly around most of the colonies on the agar surface in the plate without a noticeable pattern (Fig 1A). However, a clearly visible halo appeared around some colonies, including a control strain of *B. subtilis*, ATCC21332 (hereafter ATCC21332) that abundantly secrets surfactin [9]. We tested ~2,000 colonies over three months. Among the ~2,000 colonies, we found 18 putatively new strains that formed halos similar to or wider than ATCC21322 (Fig 1B). The entire process was estimated to be 10 to 50 times faster than the drop-collapsing assay, so could be developed into a high-throughput screening method. The manual transfer of colonies from the initial plates to clean plates was the most time-consuming and laborious, but could be automated.

### Surfactant-secreting bacteria

We chose 20 strains for identification to species level. They included the 18 strains that formed a wide, clear halo around their colony and two that did not. The 16S rRNA sequences were identical among the 18 strains with the clear zone but differed from the two strains without the zone. The 18 strains belonged to *B. subtilis*, the other two strains to *B. cereus*. For further study, we selected two of the 18 strains, ps4100 and ps4060, which respectively produced clear zones larger than or similar to ATCC21322. Although both strains abundantly secreted surfactant, the differences in the size of clear zones and growth patterns on agar plates implied that they were different strains. The size of the clear zone, not the colony, was similar but became slightly larger around ps4060 than ps4100 when they grew over a four-day period at 30°C. In contrast, the ps4060 colony was flat and about two times wider in diameter than the convoluted colony of ps4010.

### Genome features and orthologous gene phylogeny

We determined the whole genome sequences for both strains and compared them to the genomes of six other *Bacillus* species. The phylogeny of 16S rRNA genes indicated both strains

were *B. subtilis* (Fig 2A). Genome size and guanine-cytosine percentage (GC%) were more similar to *B. subtilis* than to other *Bacillus* species (Table 1). Further, the numbers of rRNA operons and tRNAs were consistently identical to *B. subtilis* and differed from other *Bacillus* species, implying that both strains were *B. subtilis*. We inquired if these strains were identical to any of the 430 *B. subtilis* strains whose genome sequences were available in GenBank. A phylogeny with 1,162 orthologous genes separated ps4100 and ps4060 into two distinct groups. Strain ps4100 was a close relative of *B. subtilis* SRCM103882 and ps4060 belonged to a group with the *B. subtilis* BAB-1 strain S2 Fig in S1 File). To further investigate their phylogenetic relationship, we built a phylogenetic tree with 780 orthologous genes in 12 selected taxa and performed bootstrap analyses (Fig 2B). The strains ps4100 and ps4060 were clearly separated into two distinct groups with strong bootstrap support (Fig 2C). *Bacillus subtilis* SRCM103882 was 26,864 nucleotides (nt) longer and contained 47 more protein-coding sequences (CDS) than ps4100 (Table 2), implying that they were different strains. Another outstanding difference between the two strains was a deleted region in the genome of SRCM103862 (Fig 3A). Conversely, *B. subtilis* BSD-2 and ps4060 differed in genome size by only 29 nt but the number of CDSs by 15 (Table 2). Although their genome organization was similar among all nine strains in this group there were spurious small-size indels across the genomes (Fig 3). In conclusion, neither strain was identical to any strain listed in GenBank.

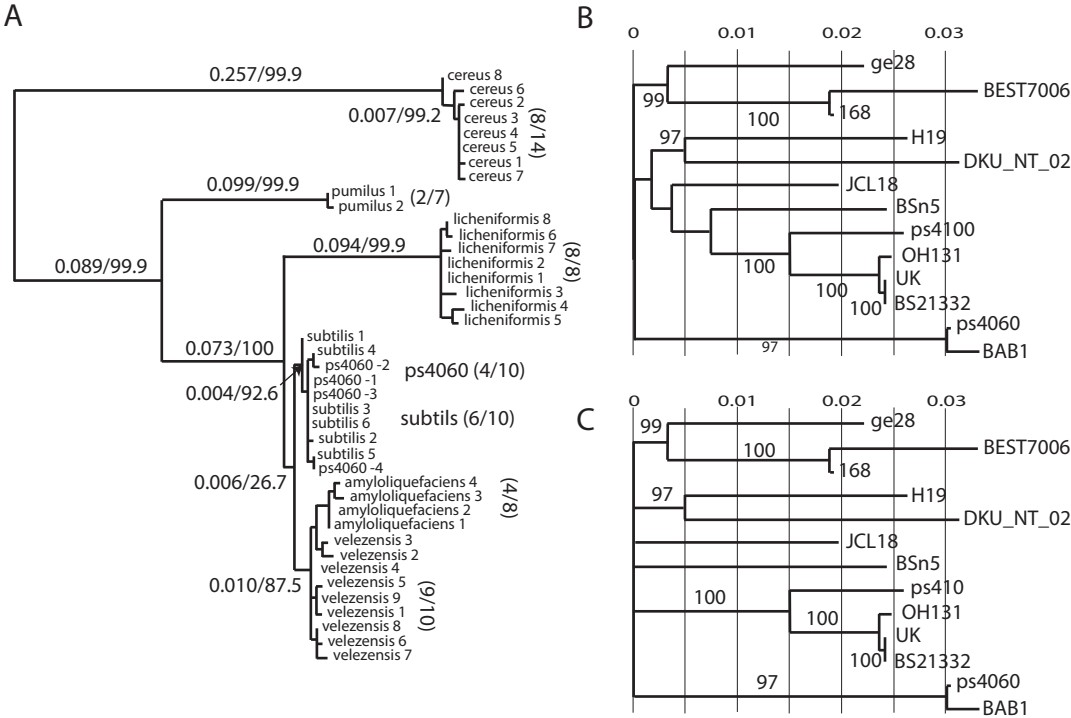

**Fig 2. Phylogenies.** A. Maximum likelihood phylogeny constructed with 16S rRNA sequences. The numbers in parenthesis indicate the total number of unique sequences and total number of rRNA operons in each *Bacillus* species. In *B. cereus*, for example, eight of fourteen 16S rRNA sequences were unique. In each strain of ps4060 and ps4100, four of ten 16S rRNA sequences were unique. These sequences were identical between ps4060 and ps4100, so only ps4060 was marked on the tree. Numbers on branches represent branch length and bootstrap values. B. Maximum likelihood phylogeny of 770 orthologous genes among 13 selected taxa. Numbers below each branch indicate bootstrap values. C. Consensus tree after collapsing nodes with bootstrap support below 50%.

**Table 1. Genome comparisons among *Bacillus* species.**

| Features | Accession # | Genome size (bp) | GC (%) | CDS | rRNA | t-RNA |
|---|---|---|---|---|---|---|
| *B. amyloliquefaciens* MT45 | NZ_CP011252.1 | 3,897,521 | 46.09 | 3,752 | 24 | 81 |
| *B. velezensis* FZB42 | NC_009725.1 | 3,918,589 | 46.48 | 3,687 | 29 | 88 |
| *B. licheniformis* DSM13T | NC_006322.1 | 4,222,645 | 46.19 | 4,223 | 24 | 72 |
| *B. pumilus* SAFR-032 | NC_009848.4 | 3,704,641 | 41.29 | 3,598 | 21 | 72 |
| *B. cereus* ATCC14579T | NC_004722.1 | 5,411,809 | 35.28 | 5,210 | 42 | 108 |
| *B. subtilis* 168 | NC_000964.3 | 4,215,606 | 43.51 | 4,328 | 30 | 86 |
| *B. subtilis* ps4100 | CP076445 | 4,106,342 | 43.78 | 4,013 | 30 | 86 |
| *B. subtilis* ps4060 | CP081458 | 4,030,808 | 43.88 | 3,875 | 30 | 86 |

CDS: number of protein-coding sequences, GC%: guanine-cytosine percentage

## Surfactin secretion

We compared the amount of surfactin secreted by the selected strains, ps4100 and ps4060, to ATCC21322 that produces about 0.1 mg/mL on complex media such as LB or nutrient broth [31]. Under the test conditions in this study, surfactin yields were respectively 0.291 and 0.177 mg/mL for strains ps4100 and ps4060, which was more than the 0.143 mg/mL yield of ATCC21322 under the same conditions (Fig 4A). A clear zone on the agar plate was larger around ps4060 than ps4100, probably because the ps4060 colony was two-times larger than ps4100 colony. Surfactin has at least four isoforms, differing in the length of their lipid tail [32]. HPLC profiles of surfactin secreted by ps4100 and ps4060 were similar to the standard surfactin (Sigma-Aldrich, Saint Louis, MO), with little variation in relative amounts among isoforms a, b, c, and d. ATCC21322 can secrete >3 mg/mL of surfactin under optimum conditions for surfactin production in chemically defined media [9]. Secretion capacity, however, was not compared among the three strains under these optimum conditions.

**Table 2. Genome comparisons among *B. subtilis* strains on the same node with ps4100 and ps4060.**

| Strain | INSDC | Size (bp) | GC% | CDS | rRNA | tRNA | φ-gene | Gene |
|---|---|---|---|---|---|---|---|---|
| ps4100 | CP076445 | 4,106,342 | 43.8 | 4,013 | 30 | 86 | 85 | 4,219 |
| SRCM103862 | CP035161.1 | 4,133,206 | 43.8 | 4,072 | 30 | 86 | 73 | 4,266 |
| MB8_B10 | CP045824.1 | 4,225,362 | 43.5 | 4,205 | 30 | 86 | 78 | 4,404 |
| IITK SM1 | CP031675.1 | 4,060,726 | 43.6 | 3,985 | 9 | 67 | 142 | 4,208 |
| R31 | CP046591.1 | 4,186,822 | 43.6 | 4,143 | 30 | 86 | 79 | 4,343 |
| SR1 | CP021985.1 | 4,093,698 | 44.2 | 3,907 | 29 | 83 | 94 | 4,118 |
| FB6-3 | CP032089.1 | 4,192,717 | 44.5 | 3,838 | 30 | 83 | 119 | 4,075 |
| AMR1 | CP050319.1 | 4,142,132 | 43.5 | 4,181 | 4 | 80 | 82 | 4,352 |
| ps4060 | CP081458 | 4,030,808 | 43.9 | 3,875 | 30 | 86 | 128 | 4,124 |
| HJ5 | CP007173.1 | 4,012,933 | 43.8 | 3,914 | 21 | 71 | 73 | 4,084 |
| BSD-2 | CP013654.1 | 4,030,837 | 43.9 | 3,890 | 30 | 86 | 110 | 4,121 |
| XF-1" | CP004019.1 | 4,061,186 | 43.9 | 3,934 | 27 | 76 | 114" | 4,156 |
| BAB-1 | CP004405.1 | 4,021,944 | 43.9 | 3,906 | 27 | 89 | 86 | 4,113 |
| SX01705" | CP022287.1 | 4,072,531 | 43.9 | 3,977 | 30 | 86 | 74 | 4,172 |
| ZD01 | CP046448.1 | 4,015,360 | 43.8 | 3,919 | 21 | 80 | 68 | 4,093 |
| UD1022 | CP011534.1 | 4,025,326 | 43.9 | 3,900 | 30 | 86 | 79 | 4,100 |
| RO-NN-1 | CP002906.1 | 4,011,949 | 43.9 | 3,889 | 30 | 86 | 82 | 4,092 |

GC%: guanine-cytosine percentage, φ-gene: pseudogene

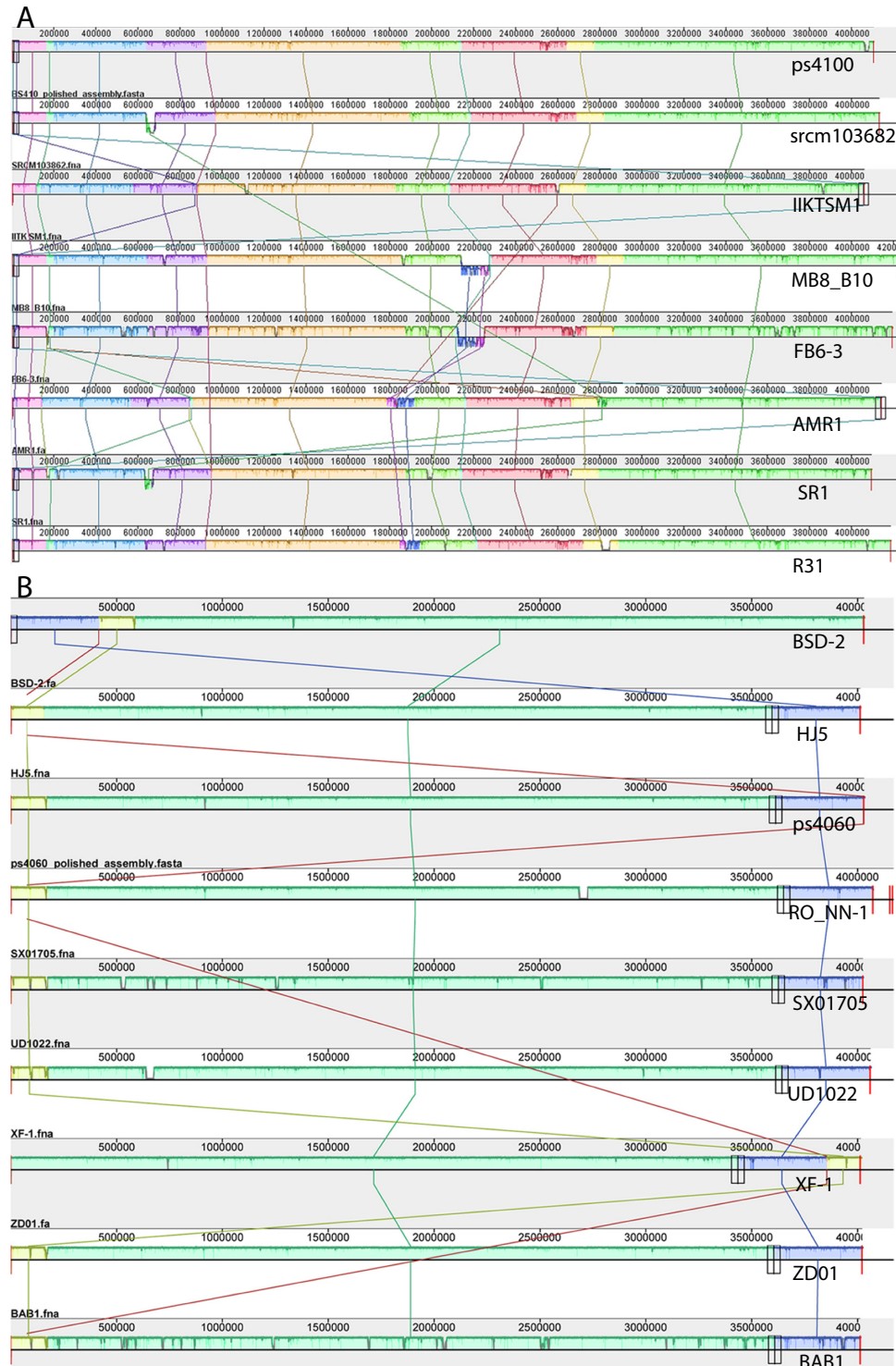

**Fig 3. Global alignment of selected genomes.** Comparison of genome structures among eight strains closely related to ps4100 (A) and nine strains close to ps4060 (B). Identically colored blocks are locally collinear regions of multiple gene sequences without rearrangement of homologous genes. Each line connects a set of collinear blocks. Colored blocks below the centerline indicate inverted sequences.

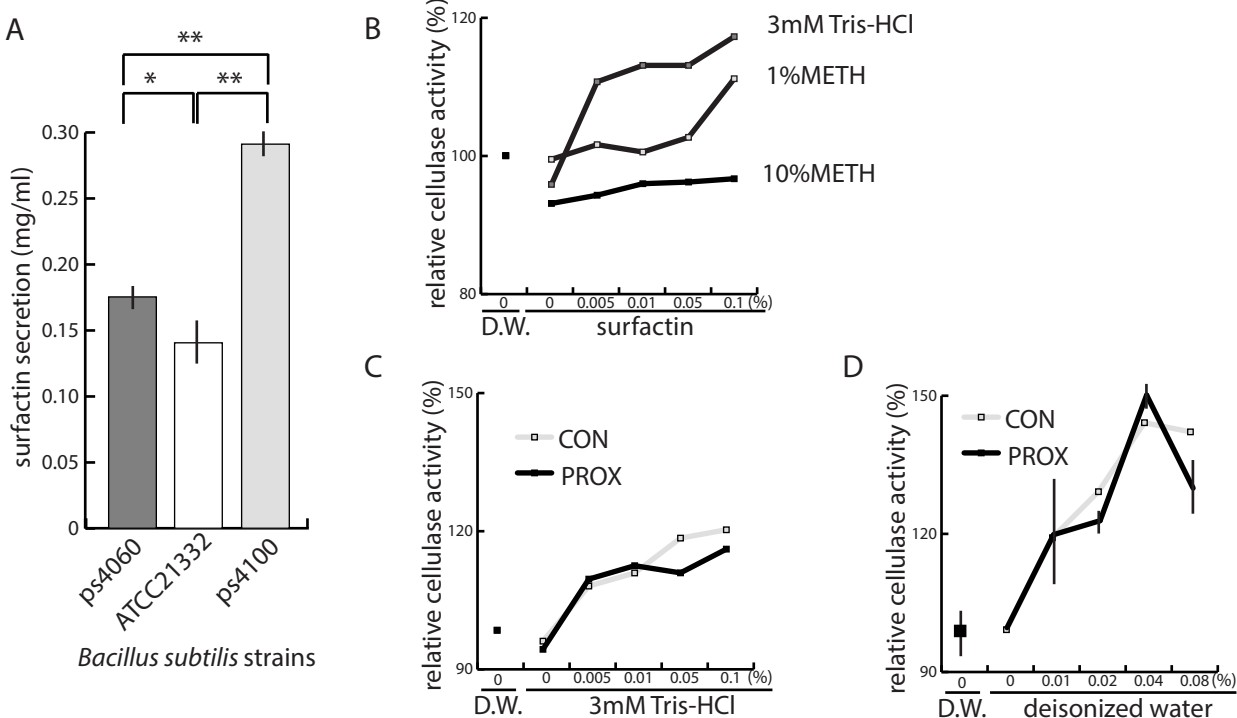

**Fig 4. Surfactin production by two novel *Bacillus subtilis* strains compared to ATCC21322 and the effect of surfactin on cellulase activity.** A. Comparison of surfactin secreted in a 50-mL medium in triplicates at 30°C. B. Effect of surfactin on cellulase activity in the presence of 3 mM Tris-HCl (pH 8.0), 1% methanol, 10% methanol. C and D. Effects of two sources of surfactin, Sigma-Aldrich (CON) and Proxenrem (PROX), on cellulase activity in the presence of 3mM Tris-HCl and deionized water. Control surfactin was purchased from Sigma-Aldrich, St. Louis, MO, USA. The x-axes in B, C, and D, show surfactin concentrations. Bars show standard deviations. Single (*) and double (**) asterisks respectively indicate *p*-values less than 0.05 and 0.001 by Student-tests.

## Surfactin enhances cellulase activity

We investigated the effects of surfactin on cellulase activity, which is probably important for the digestion of plant-based diets. Cellulase releases glucose from cellulose and several surfactants enhance the enzyme's activity [33]. In this study, cellulase activity decreased up to 10% in the presence of 10% methanol or Tris-HCl buffer in the reaction (Fig 4B–4D). Surfactin overcame the deleterious effects of both methanol and Tris-HCl, enhancing cellulase activity by about 20% (Fig 4B and 4C). Surfactin in water without methanol enhanced enzyme activity by up to 50% (Fig 4D). Overall, the presence of surfactin in the reaction mixture resulted in significantly more glucose released from cellulose.

## Surfactin suppressed inducible NO synthetase and NO production *in vitro*

Pathogen infection and stresses such as weaning and sudden weather changes can induce inflammation and slow the growth of mammals [1, 34, 35]. We tested whether surfactin suppressed inflammation *in vitro*, possibly enhancing growth rates. We used four major isoforms of surfactin a, b, c, and d fractionated (S1 Fig in S1 File). None of the four major isoforms affected cell viability up to a 50 μM concentration (Fig 5A). Each isoform, however, inhibited nitric oxide (NO) production in macrophage RAW 264.7 cells activated by lipopolysaccharides (LPS) in a concentration-dependent manner. The IC$_{50}$ concentration of isoforms a, b, c, and d were 17.7, 19.5, 20.5, and 20.4 μM, respectively (Fig 5B). The inhibitory effect was strong for each isoform with little difference among them. To determine the cause of decreased NO, we

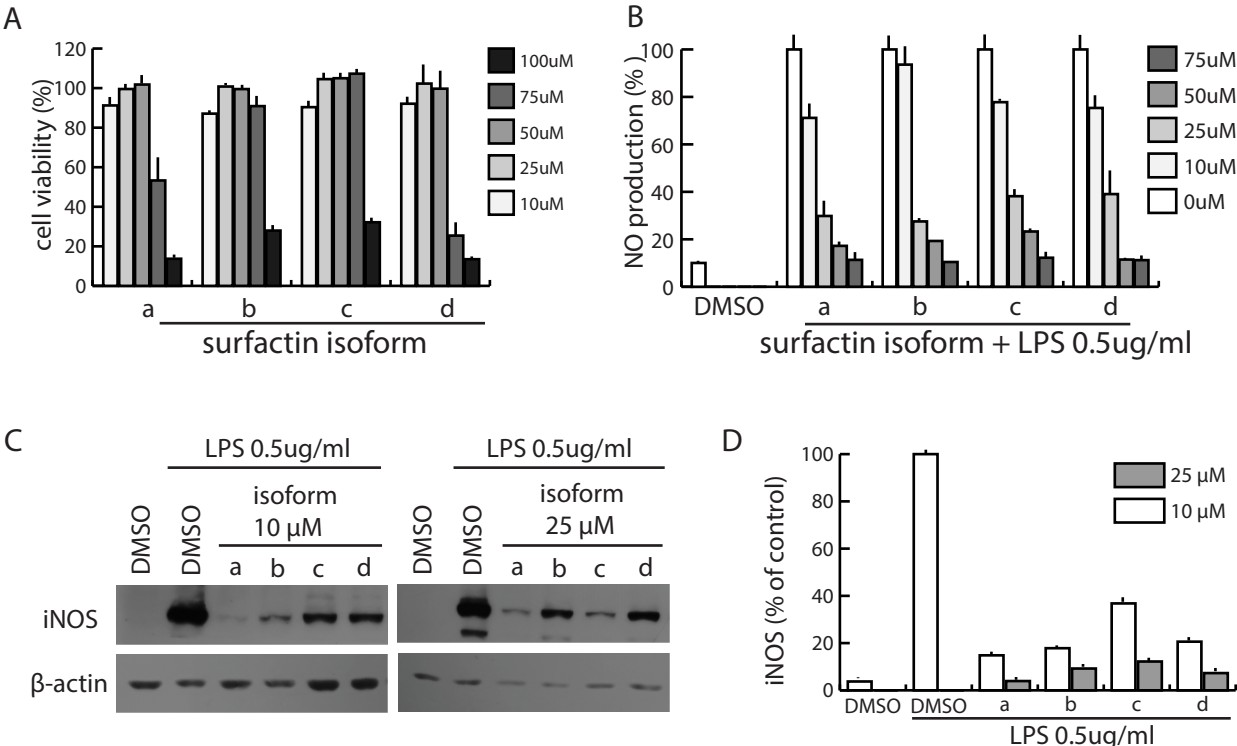

**Fig 5. Suppression of nitric oxide (NO) production by surfactin.** A. Effect of surfactin isoforms a, b, c, and d on the viability of macrophage cells, RAW 264.7. B. Effect of surfactin on NO production. C. Western blot results showing suppression of the inducible NO synthetase (iNOS) by each surfactin isoform. Lipopolysaccharide (LPS) dissolved in dimethyl sulfoxide (DMSO) is an iNOS inducer. D. Chart showing relative amounts of iNOS. Boxes and bars represent mean and standard deviations.

analyzed the effect of each surfactin isoform on the expression of inducible NO synthetase (iNOS). This enzyme is needed to produce NO, an intracellular inflammatory mediator [36]. Ten and 25 μM of each derivative respectively caused an 80 and 90% decrease in the expression of inducible NO synthetase (iNOS, Fig 5C and 5D). The iNOS expression pattern implied that surfactin suppressed iNOS, followed by a decrease in the NO production that mediates inflammation *in vivo*.

## Absence of antibiotic resistance

There were no antibiotic-resistant genes in the genome of either strain. Accordingly, both strains were highly susceptible to all six antibiotics tested in this study (Fig 6). Thus, these strains are suitable probiotic candidates according to the regulation set by the Korean Food and Drug Administration, a guideline concerning the assessment of bacterial susceptibility to antimicrobials of human and veterinary importance [37].

## Discussion

*Bacillus subtilis* is generally recognized as a safe organism [38]. It is included in traditionally fermented foods, such as Korean Cheonggukjang and Japanese Natto that humans have consumed for hundreds of years. It is also safe for animal consumption. The secretion of surfactant was used in our study as a biomarker to further screen *B. subtilis* for potent probiotic candidates. Biosurfactant-producing bacteria can be screened using several assays, such as drop collapse, emulsification, tensinometric evaluation, fluorescence detection, and the oil-

ps4060

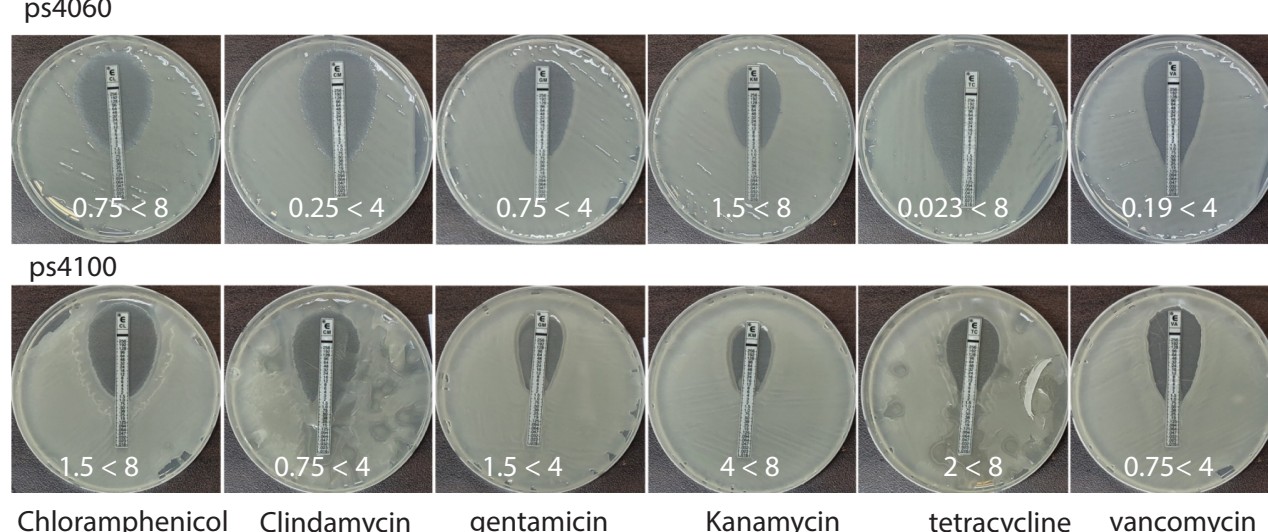

ps4100

| Chloramphenicol | Clindamycin | gentamicin | Kanamycin | tetracycline | vancomycin |

**Fig 6. Susceptibility to six antibiotics.** In each frame, the value on the left is the minimal inhibitory concentration (MIC) measured in this study; on the right is the acceptable MIC designated by the Korean Food and Drug Administration. The experiment was conducted using the Epsilometer test strip (E-test). Unit: µg/mL.

misting method [10, 39, 40]. We chose the oil-misting method to assess *B. subtilis* strains, as it was suitable for a high-throughput screening of surfactant-producing bacterial strains. After manually screening ~2,000 colonies, we selected two strains, ps4100 and ps4060, mainly because they secreted more surfactin than *B. subtilis* ATCC21332 (Fig 1). They were novel strains (Figs 2 and 3), susceptible to antibiotics (Fig 6), thus eliminating concerns over transferring antibiotic resistance from these strains and suitable as probiotic candidates.

*Bacillus* spp. are potentially good probiotics because they form spores that survive, germinate, and temporarily persist as vegetative cells in gastrointestinal tracts [3, 4, 41]. Although several strains have been developed as animal growth promoters, ps4100 and ps4060 abundantly secret surfactin and could be excellent probiotics for several reasons. The structure and amphiphilic nature of surfactin could disrupt cell membranes and inactivate or kill various microorganisms, including bacteria, viruses, fungi, and mycoplasma [42]. Although the antimicrobial efficacy of surfactin alone may not be strong, it acts synergistically with other surfactants and antimicrobial agents [43, 44]. If the new *Bacillus* strains secret other antimicrobial bacteriocins, their efficacy would also be enhanced by surfactin [43]. Its antifungal activities could protect animals by suppressing *Aspergillus* and *Fusarium*, which produce potent mycotoxins such as aflatoxin, trichothecenes, and zearalenone [43, 45]. Up to 90% of oral-fed *Bacillus* spores germinate and multiply as vegetative cells in the gastrointestinal tract of pigs [4]. Like antibiotics, expression of surfactin in the gastrointestinal tract would suppress the growth of disease-causing microbes and acts as animal growth promoters at a low concentration. It is notable that ps4100 and ps4060 produced about twice the surfactin as strain ATCC21332, which produces ~100 µg/mL in a nutrient-rich environment and up to 600 mg/mL under optimum condition [9]. The strain ATCC21332 arguably secretes the highest amount of surfactin among known *B. subtilis*. The gastrointestinal tract, however, provides suboptimal conditions for *Bacillus* to secrete surfactin due to its high-nutrient and low-oxygen levels. Thus, the ability of ps4060 and ps4100 to secrete a large amount surfactin in a nutrient-rich environment would be an important trait against pathogenic microorganisms.

There was additional evidence that surfactin-secreting *Bacillus* spp. might be better growth promoters than antibiotics. In our study, surfactin increased cellulase activity up to 50% *in vitro* (Fig 4). Cellulose is the primary component of plant cell walls, the most abundant organic molecule on earth, and the least expensive in animal diets. Efficient digestion could benefit substantially because glucose released from cellulose could reduce the use of corn and wheat as sources of starch in animal diets. It is unclear whether a 50% increase in cellulase activity is sufficient to produce a tangible effect on diet efficiency. Cellulases are generally one-to-two orders less reactive than other enzymes, such as amylases and proteases [46]. The effects of surfactin on cellulases could be substantial if the quantity of cellulase or affinity for its substrate is over ten-times higher than other enzymes. It is worth investigating if surfactin enhances the activities of other nutrient-digesting enzymes like the hemicellulases and pectin-digesting enzymes.

Surfactin may affect animal immune systems. The iNOS expression pattern *in vitro* implies that even at low concentrations surfactin suppresses the expression of iNOS, followed by a decrease in NO production and inflammation (Fig 5). It has a positively charged peptide loop of seven amino acids with a long hydrophobic fatty acid chain [32]. This amphiphilic structure enables it to interact with LPS that activates NO production and subsequent inflammation [47]. Lipopolysaccharide is one of two major products of gram negative bacteria among gut microbes that enter the blood stream and result in inflammation [48]. Current data are insufficient to address whether surfactin suppresses inflammation *in vivo* by affecting inflammation-associated genes and pathways.

In initial efficacy tests during animal growth studies, the mixture of ps4060 and ps4100 enhanced the productivity of pigs and reduced noxious gases in fecal discharges [49]. Encouragingly, novel *Bacillus* strains abundantly secreting surfactin *in vitro* appeared to be good probiotic candidates and that high-throughput screening was possible.

## Supporting information

**S1 File.**
(DOCX)

**S2 File.**
(PNG)

**S3 File.**
(PNG)

## Acknowledgments

We thank Heoyksu Kweon and Fred Brooks for assisting in enzyme assays and editing the manuscript.

## Author Contributions

**Conceptualization:** Jong Seok Ahn, Yangrae Cho.

**Data curation:** Dongho Lee, Taehoon Oh, Yangrae Cho.

**Formal analysis:** Dongho Lee, Bongseok Kang, Yangrae Cho.

**Funding acquisition:** Yangrae Cho.

**Investigation:** Taehoon Oh, Bongseok Kang.

**Methodology:** Yangrae Cho.

**Software:** Dongho Lee.

**Supervision:** Jong Seok Ahn, Yangrae Cho.

**Validation:** Yangrae Cho.

**Visualization:** Taehoon Oh, Yangrae Cho.

**Writing – original draft:** Dongho Lee, Yangrae Cho.

**Writing – review & editing:** Jong Seok Ahn, Yangrae Cho.

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
