## [Decision Letter · Decision Letter 0]

7 Sep 2022

PONE-D-22-21217Throughput screening of Bacillus subtilis strains that abundantly secrete surfactin in vitro identifies effective probiotic candidatesPLOS ONE

Dear Dr. Cho,

Thank you for submitting your manuscript to PLOS ONE. After careful consideration, we feel that it has merit but does not fully meet PLOS ONE’s publication criteria as it currently stands. Therefore, we invite you to submit a revised version of the manuscript that addresses the points raised during the review process.

We look forward to receiving your revised manuscript.

Kind regards,

Joseph Selvin, Ph.D

Academic Editor

PLOS ONE

Journal Requirements:

"This research was partly supported by a goodwill program of the Korean Steel Corporation to YC, R&D fund of the Cheongbuk-do and Cheongju-si local government to YC, and the Technology development Program (S3089057) of the Ministry of SMEs and Startups (MSS, Korea) to YC."

"This research was partly supported by a good will program of the Korean Steel Corporation, R&D fund of the Cheongbuk-do and Cheongju-si local government, and the Technology development Program (S3089057) of the Ministry of SMEs and Startups (MSS, Korea)."

"This research was partly supported by a goodwill program of the Korean Steel Corporation to YC, R&D fund of the Cheongbuk-do and Cheongju-si local government to YC, and the Technology development Program (S3089057) of the Ministry of SMEs and Startups (MSS, Korea) to YC."

"YC owns the startup company, ProxEnrem Inc., where he identified the Bacillus strains as candidates for animal diet supplements. However, we do not anticipate an immediate financial benefit from this publication. It may be easier to acquire a permit to register the strains for animal diet supplements. This work was mostly supported by the Korean Steel Corporation (KSC) as a social responsibility by indirectly assisting technology development and a reduction in environmental pollution to compensate for the inevitable production of CO2 during the company’s operation. KSC will not have a direct financial benefit from this publication and they influenced neither study design nor data interpretation."

"YC owns the startup company, ProxEnrem Inc., where he identified the Bacillus strains as candidates for animal diet supplements. However, we do not anticipate an immediate financial benefit from this publication. It may be easier to acquire a permit to register the strains for animal diet supplements. This work was mostly supported by the Korean Steel Corporation (KSC) as a social responsibility by indirectly assisting technology development and a reduction in environmental pollution to compensate for the inevitable production of CO2 during the company’s operation. KSC will not have a direct financial benefit from this publication and they influenced neither study design nor data interpretation."

We note that you received funding from a commercial source: Korean Steel Corporation (KSC)

Within this Competing Interests Statement, please confirm that this does not alter your adherence to all PLOS ONE policies on sharing data and materials by including the following statement: ""This does not alter our adherence to PLOS ONE policies on sharing data and materials.” (as detailed online in our guide for authors http://journals.plos.org/plosone/s/competing-interests).  If there are restrictions on sharing of data and/or materials, please state these. Please note that we cannot proceed with consideration of your article until this information has been declared. 

7. Please upload a new copy of Figure 3 as the detail is not clear. Please follow the link for more information: https://blogs.plos.org/plos/2019/06/looking-good-tips-for-creating-your-plos-figures-graphics/
https://blogs.plos.org/plos/2019/06/looking-good-tips-for-creating-your-plos-figures-graphics/

Reviewers' comments:

Reviewer's Responses to Questions

**Comments to the Author**

1. Is the manuscript technically sound, and do the data support the conclusions?

Reviewer #1: Yes

Reviewer #2: Partly

Reviewer #3: Yes

2. Has the statistical analysis been performed appropriately and rigorously? 

Reviewer #1: N/A

Reviewer #2: I Don't Know

Reviewer #3: N/A

3. Have the authors made all data underlying the findings in their manuscript fully available?

Reviewer #1: Yes

Reviewer #2: No

Reviewer #3: Yes

4. Is the manuscript presented in an intelligible fashion and written in standard English?

Reviewer #1: Yes

Reviewer #2: Yes

Reviewer #3: Yes

5. Review Comments to the Author

Reviewer #1: This manuscript describes an alternative antibiotic used in animal farming.

Although the research is interesting in that the results show some beneficial probiotic properties such as surfactin, the manuscript lacks some vital information as follows:

Abstract:

Page 2, line 29, line 34: “LB”, "IC" Provide the full name and then abbreviated.

Material and Methods:

Page 10, line 156-164: The surfactin production medium detail and procedure; please, prove with reference (s).

Page 11-14, line 173-221: The cellulase activity, NO production, and iNOS protein measurement should be proven with reference (s).

Page 15: For antibiotics resistance tests, the authors should state the interpretation of the test and the reference.

Results:

Page 20, line 299-300: The scale of surfactin yields in Figure 4A did not match the result reported in lines 299-300. Please, rescale for the matching.

Discussion:

Page 23, line 356: "Bacillus sublitis is generally as safe..." This statement should be proven with reference.

Page 24, line 368-370: This statement citation is missing in the reference section. Please, provide the searchable source in the reference section.

Page 25, line 383-385: Please, prove this statement with reference. In addition, the authors should state the significant finding about surfactin production in this study,

such as how the selected isolates produced a large or small amount of surfactin compared to other surfactin-producing bacteria in previous studies.

Page 27, line 413-414: The statement citation is missing in the reference section.

Figure 3: Please, describe the result from the figure in more detail, and the program use should be mentioned.

Statistical analysis should be added.

Reviewer #2: The manuscript deals with the characterization of two novel B. subtilis strains identified through a screening based on the amount of surfactin secreted. Authors see correlation between the amount of surfactin and induction of cellulase activity (in vitro), reduction of NO production and reduction of nitric oxide synthetase in a cell line. Finally, they showed that the strains are sensitive to several antibiotics.

The paper is interesting and is generally well written with some localized problems that have been specifically addressed below.

In general, M&M and also figure legends are not always detailed and require revision. Also, the presentation of the data in the Results section requires revision.

No one reads the Methods before the Results; the latter should thus be self-standing and logically presented. Therefore, my request is to move the sentence 89-90 (“To discover surfactant-producing bacteria among heat-tolerant microorganisms, a large number of microbial colonies were screened”) at the beginning of the Results section, where it plays a fundamental role. Also, add the source of the samples that were screened (did they come from a personal collection? Or from a laboratory library? How were sample chosen?) at the beginning of the Results section.

Same fate for the sentence 202-204: “To determine the cause of decreased NO, we analyzed the effect of each surfactin isomer on the expression of inducible NO synthetase (iNOS). This enzyme is needed to produce NO, an intracellular inflammatory mediator.” Its correct location is in the Results section.

More importantly, some controls are missing from the experiments. After fractionation, was surfactin used in the following experiments as such or was it dried and resuspended? If so, in which buffer? At which concentration? Have you resuspended a surfactin-void HPLC fraction in the same buffer to distinguish the effect of buffer from the one of surfactin?

This is very important for the experiments shown in Figs. 4 and 5. Sigma surfactin is normally resuspended in NaOH and this might cause a drastic change in the pH of the enzymatic reaction (cellulases are sensitive to pH changes). Therefore, the detailed protocol of surfactin production and use in each experiment is required. Have Authors tried to add an equal amount of surfactin resuspension buffer to the cellulases reactions to identify its effect?

Control experiments with pure surfactin from a commercial source should be shown as positive controls. Besides, you could use a non-producer strain processed in parallel with your selected ones as negative control in the different assays. Indeed, the surfactin purification protocol is quite vague and B. subtilis produces so many additional secreted molecules that could be responsible for the activities shown.

Furthermore, fig. 4E states that Tris was used and not water (line 321 and figure legend).

It would be interesting to see the HPLC profile of surfactin isoforms. In line 32 it is said there are at least 4 isoforms; in line 184 it is stated that surfactin was fractionated into 4 isoforms and therefore it sounds odd to read “4 of the individual isoforms” in lines 33-34. Could you please show the data and, if it is the case, specify whether the 4 chosen are the “major isoforms”?

Also important is the recurrent quotation -and even a part of discussion – of a putatively accepted manuscript. Please remove the information that refers to unpublished data from the abstract (lines 21-23).

Since it is stated that the manuscript has been accepted for publication (lines 370, 414), please quote it as a reference in the bibliography with the full list of authors and the journal (example at https://www.nlm.nih.gov/bsd/uniform_requirements.html#unpublished: “Tian D, Araki H, Stahl E, Bergelson J, Kreitman M. Signature of balancing selection in Arabidopsis. Proc Natl Acad Sci U S A. Forthcoming 2002.”). However, remove the paragraph from line 413 to 423 as it pertains to the discussion of other results that have nothing to do with the current manuscript.

Minor comments:

Authors should describe how IC50 was calculated.

77-78: Could you clarify for readers (i.e., in the text) why “surfactant-secreting strains because would not trigger the emergence of resistant pathogens even after extensive use”? Which could be the other characteristics that can be screened that do trigger resistance, besides antibiotic production?

Reference 2: wrong journal name.

56: gas emission. Could you please specify whether the reduction is supposedly derived from lower methane production from animals or less carbon dioxide from land use or reduced nitrous oxide from manure and slurry management? Please, quote a reference for this strong statement.

65: “their effect” -> their probiotic effect

98: Bacillus subtilis should be in italics

115: cell (not plural)

116: reference 9 is wrong, at least in this position

122: substitute “Their” with “The “

124: the accession number CP081458 does not provide any result in GenBank. Please check it.

124-25: Please provide the link to the “Korea Microbe Collection Center” and specific links to the accession numbers KACC81161BP 125 and KACC81159BP. It was impossible to find them through google search.

129-133: paragraph unclear for several reasons. Using the past tense and the singular verb (There were 10 operons in B. subtilis genome) I do not understand whether the information concerns one of the two strains newly described or 168 or else. The use of the past tense made me think that the description refers to the new strains, but then it should be plural… also, the rest of the paragraph is obscure. “All” means many. Which are the genomes that are under analyses here? Please, rephrase it.

135: add references to the list and substitute with numbers.

141: “two unknown strains”. Which is the source of these strains?

150: representing?

161: incubated?

165: 5 µM refers to column diameter? Please specify

177: significance of ΔΔ symbol should be given.

216: Which is the source of the antibodies for iNOS and actin?

237: “We tested ~2,000 colonies for three months” means that the screening procedure took three months or it means that you repeated the same screening over and over for 90 days?

257-58: Could you show the “differences in the size of clear zones and growth patterns” for the two selected strains?

264: than TO other

268-270: “…each was respectively the closest relative to the B. subtilis SRCM103882 and BAB-1 groups (Fig 2B).” please rephrase.

272: subtilis

278: please, enrich legend to Figure with more details. You should explain why each strain appears with different numbering (does it refers to different rRNA operons?). Define which method was used for the tree in A (maximum likelihood or others?)

312: legend to fig. 4E does not correspond to what shown in the figure.

327 “thereby possibly enhancing” instead of which

329 please move “activated” after “cells”. Please account for the presence of LPS in fig. 5A & B and in the corresponding legend.

347: odd sentence “recommendations based on a guideline concerning…”.

350: please change the title to Fig. 6 to a sentence referring to the scientific meaning of it (not to the method)

357: natto

358: have been?

366: prone to antibiotics?

367-368: “The possibility appeared to be very low to transfer antibiotic resistance from these strains to animals or farm workers”??? It was just stated that the strains do not contain resistance genes. Is this sentence meaningful?

Reviewer #3: This manuscript describes how two new Bacillus subtilis strains were selected for screening probiotic candidates, taking into account surfactin secretion as a marker in vitro, and identified by genome sequencing. In addition, four surfactin isoforms are described, in which the cellulase activity and the mean maximal inhibitory concentration (IC50) were determined; as well as the susceptibility to several antibiotics of both strains. The article is also supported by another in vivo test in pigs (L21-23), which is reported to be in press, according to what is written by the authors in L370 and 414.

The topic is interesting, and the text is well structured and written. However, the authors can find here some suggestions and corrections. My main concern is about referencing properly to the publication related to the in vivo test. This study is under review or has already been accepted for publication. Where and who are the authors? Please, clarify and add this reference to the references section, detailing the year in the in-text citations.

Authors wrote (L103-104): “Halo quality was arbitrarily assessed by clarity, transparency, and evenness of oil layer in the clear zone.” Arbitrarily? Not very scientific, is it? How were clarity, transparency and uniformity evaluated? For example, using an increasing subjective 3- or 4-point scale.

L120: “This process included consensus polishing with Quiver”. Please, add a citation.

L135: “(van Berkum et al., 2003; Yabuki et al., 2014)”. Please, pay attention since these citations are missing in the references section.

L166: “(Corning, Germany)” Which is the city? Please, be consistent with other resources cited.

L197 “(Molecular Devices, Spectra Max 190)” Which is the location of company? See comment above.

L220 “(Thermo Fisher, Rochford, USA)” Sometimes the U.S. city and state of the company are listed, but not this time. However, other times the country is not listed. Please, be consistent according to the journal guidelines. Check it throughout the M&M section.

L247: “Black circles indicate two colonies with larger, clearer halos than the control.” Black? They are red in the draft.

L263: “Genome size and GC% were more similar to B. subtilis than other Bacillus species (Table 1)” Guanine-cytosine percentage? Acronyms should be described the first time they are used in the text (they should also be detailed in the tables and/or figures).

Fig 2C and 2D are not mentioned in the text. Moreover, they both look identical. What are they used for?

L298: “Under the test conditions in this study, surfactin yields were respectively 0.3 and 0.2 mg/mL for strains ps4100 and ps4060, exceeding the 0.15 mg/mL yield of ATCC21322 mg/mL under the same conditions (Fig 4A)” Both strains cannot exceed 0.15 mg/mL because there is a difference of 0.1 mg/mL between these two lines. Rewrite this sentence accordingly. In addition, according to the figure 4A (with bars and lines), there appears to be some variation in the quantity produced. How was the difference between the amount of surfactin produced between two strains evaluated (statistical comparison to assess whether it was different from zero)? Figures should be self-explanatory. On the other, using the same range or amplitude of the y-axis of Figures 4B-E would improve the visual comparison.

L350-353 and Figure 6. What were the units? A number is always followed by its unit.

L400: “The effects of surfactin on cellulases could be substantial if the quantity or Km is over ten-times higher than other enzymes.” Km??? Please, clarify.

References: Sometimes the name of the journal is abbreviated (e.g. Reference number 9, 17, etc.), sometimes it is not. Please, use journal names in ISO4 format. Check it throughout the section. In the reference 2, the name of journal is only Animals (remove : an open access journal from MDPI).

6. PLOS authors have the option to publish the peer review history of their article (what does this mean?). If published, this will include your full peer review and any attached files.

Reviewer #1: No

Reviewer #2: No

Reviewer #3: No

---

## [Author Response · Author response to Decision Letter 0]

19 Oct 2022

PONE-D-22-21217

Throughput screening of Bacillus subtilis strains that abundantly secrete surfactin in vitro identifies effective probiotic candidates

Reviewers' comments:

Reviewer's Responses to Questions

Comments to the Author

Reviewer #1: This manuscript describes an alternative antibiotic used in animal farming.

Although the research is interesting in that the results show some beneficial probiotic properties such as surfactin, the manuscript lacks some vital information as follows:

Abstract:

Page 2, line 29, line 34: “LB”, "IC" Provide the full name and then abbreviated.

our response) LB = Luria Berani, IC50 = the half-maximal inhibitory concentration 

Material and Methods:

Page 10, line 156-164: The surfactin production medium detail and procedure; please, prove with reference (s). 

our response) The medium composition was determined based on our fermentation experience and three publications. The publications were referenced appropriately.

Page 11-14, line 173-221: The cellulase activity, NO production, and iNOS protein measurement should be proven with reference (s). 

our response) We elaborated this section with references: 

1) Cellulase activity section (L182-188): Surfactin was completely dried and dissolved in methanol (10 mg/mL). The resulting solution was subsequently diluted with deionized water or an appropriate buffer as described for each experiment. Surfactin solution was added to the reaction mixture and cellulase activity measured (Worthington, 1988). The cellulose substrate was dissolved in a large volume of 50 mM acetate buffer (pH 5.0). An addition of deionized water without buffer, or a small amount of 1 M Tris-HCl buffer (pH 8.0) to make a 3 mM concentration, did not affect the pH of the reaction mixture. 

2) NO production, iNOS protein measurement section (L199-204): We fractionated four major isoforms of surfactin a, b, c, and d (S1 Fig) from the total surfactin dissolved in 99.99% methanol by a preparative HPLC system (Agilent Technology, Santa Clara, CA, USA). The fractionated isoforms were completely dried in a SpeedVac vacuum drier (Eppendorf, Hamburg, Germany), and then dissolved in dimethyl sulfoxide (DMSO), which was effective regardless of pH (Abdel-Mawgoud et al., 2008). We used DMSO alone in a negative control experiment. 

3) References used in this section:

4) Park JE, Cuong TD, Hung TM, Lee I, Na M, Kim JC, et al. Alkaloids from Chelidonium majus and their inhibitory effects on LPS-induced NO production in RAW264.7 cells. Bioorg Med Chem Lett. 2011;21(23):6960-3.

5) Banskota AH, Stefanova R, Sperker S, Lall SP, Craigie JS, Hafting JT, et al. Polar lipids from the marine macroalga Palmaria palmata inhibit lipopolysaccharide-induced nitric oxide production in RAW264.7 macrophage cells. Phytochemistry. 2014;101:101-8.

6) Lin Q-Y, Jin L-J, Cao Z-H, Xu Y-P. Inhibition of inducible nitric oxide synthase by Acanthopanax senticosus extract in RAW264.7 macrophages. Journal of Ethnopharmacology. 2008;118(2):231-6.

Page 15: For antibiotics resistance tests, the authors should state the interpretation of the test and the reference. 

our response) After incubation, an inhibition zone of bacterial growth forms as an ellipse that intersects the MIC reading scale on the ug/ml unit (Goldstein et al., 2007).

Results:

Page 20, line 299-300: The scale of surfactin yields in Figure 4A did not match the result reported in lines 299-300. Please, rescale for the matching. 

our response) We changed the unit from mg/L to mg/ml in Figure 4

Discussion:

Page 23, line 356: "Bacillus sublitis is generally as safe..." This statement should be proven with reference. 

our response) a reference (Elshaghabee et al., 2017) was added.

Page 24, line 368-370: This statement citation is missing in the reference section. Please, provide the searchable source in the reference section.

This sentence was removed, but the citation was included in another section. 

Page 25, line 383-385: Please, prove this statement with reference. In addition, the authors should state the significant finding about surfactin production in this study, such as how the selected isolates produced a large or small amount of surfactin compared to other surfactin-producing bacteria in previous studies.

our response) Line 437-444. We added the sentence: Up to 90% of oral-fed Bacillus spores germinate and multiply as vegetative cells in the gastrointestinal tract of pig (Leser et al., 2008). Like antibiotics, expression of surfactin in the gastrointestinal tract would suppress the growth of disease-causing microbes and acts as animal growth promoters at a low concentration. It is notable that ps4100 and ps4060 produced about twice the surfactin, or slightly more, than ATCC21332, which produces ~100ug/ml in a nutrient-rich environment and up to 600 mg/ml under optimum conditions (Chen et al., 2015). The strain ATCC21332 arguably secretes the highest amount of surfactin among known B. subtilis. 

Page 27, line 413-414: The statement citation is missing in the reference section.

our response) This sentence was removed, but the citation was included in another section. 

Figure 3: Please, describe the result from the figure in more detail, and the program use should be mentioned. 

our response) Figure 3. Global alignment of selected genomes. Comparison of genome structures among eight strains closely related to ps4100 (A) and nine strains close to ps4060 (B). Identically colored blocks are locally collinear regions of multiple gene sequences without rearrangement of homologous genes. Each line connects a set of collinear blocks. Colored blocks below the centerline indicate inverted sequences. 

Statistical analysis should be added.

our response) Results of the statistical analysis were included in Figure 4 A, but the analytical method was not described in the M&M section. The test was a simple Student-t test and the results were obvious without the analyses, as indicated by reviewer 2.

Reviewer #2: The manuscript deals with the characterization of two novel B. subtilis strains identified through a screening based on the amount of surfactin secreted. Authors see correlation between the amount of surfactin and induction of cellulase activity (in vitro), reduction of NO production and reduction of nitric oxide synthetase in a cell line. Finally, they showed that the strains are sensitive to several antibiotics.

The paper is interesting and is generally well written with some localized problems that have been specifically addressed below.

In general, M&M and also figure legends are not always detailed and require revision. Also, the presentation of the data in the Results section requires revision.

No one reads the Methods before the Results; the latter should thus be self-standing and logically presented. Therefore, my request is to move the sentence 89-90 (“To discover surfactant-producing bacteria among heat-tolerant microorganisms, a large number of microbial colonies were screened”) at the beginning of the Results section, where it plays a fundamental role. Also, add the source of the samples that were screened (did they come from a personal collection? Or from a laboratory library? How were sample chosen?) at the beginning of the Results section.

Same fate for the sentence 202-204: “To determine the cause of decreased NO, we analyzed the effect of each surfactin isomer on the expression of inducible NO synthetase (iNOS). This enzyme is needed to produce NO, an intracellular inflammatory mediator.” Its correct location is in the Results section.

our response) We kept the sentence 89-90 in the M&M section and added two new sentences at the beginning of the Results section. Line 255-257. A large number of heat-tolerant bacteria from a personal collection of insect feces and soils from remote mountains were screened to discover novel strains of surfactant-producing Bacillus species . 

sentence 202-204. This sentence was move to the Results (L384-386) section and this reference was added. “To determine the cause of decreased NO, we analyzed the effect of each surfactin isomer on the expression of inducible NO synthetase (iNOS). This enzyme is needed to produce NO, an intracellular inflammatory mediator (Bogdan, 2015).

More importantly, some controls are missing from the experiments. After fractionation, was surfactin used in the following experiments as such or was it dried and resuspended? If so, in which buffer? At which concentration? Have you resuspended a surfactin-void HPLC fraction in the same buffer to distinguish the effect of buffer from the one of surfactin?

This is very important for the experiments shown in Figs. 4 and 5. Sigma surfactin is normally resuspended in NaOH and this might cause a drastic change in the pH of the enzymatic reaction (cellulases are sensitive to pH changes). Therefore, the detailed protocol of surfactin production and use in each experiment is required. Have Authors tried to add an equal amount of surfactin resuspension buffer to the cellulases reactions to identify its effect?

Control experiments with pure surfactin from a commercial source should be shown as positive controls. Besides, you could use a non-producer strain processed in parallel with your selected ones as negative control in the different assays. Indeed, the surfactin purification protocol is quite vague and B. subtilis produces so many additional secreted molecules that could be responsible for the activities shown.

our response) (NaOH was not used to resuspend the surfactin) We elaborated on the section of the cellulase activity experiment and Figure 4: Line 182-188. Surfactin was completely dried and dissolved in methanol (10 mg/mL). The resulting solution was subsequently diluted with deionized water or an appropriate buffer as described for each experiment. Surfactin solution was added to the reaction mixture and cellulase activity measured (Worthington, 1988). The cellulose substrate was dissolved in a large volume of 50 mM acetate buffer (pH 5.0). An addition of deionized water without buffer, or a small amount of 1 M Tris-HCl buffer (pH 8.0) to make a 3 mM concentration, did not affect the pH of the reaction mixture. 

For the NO-related experiment, we dissolved surfactin in DMSO (Line 199-204). We fractionated four major isoforms of surfactin a, b, c, and d (S1 Fig) from the total surfactin dissolved in 99.99% methanol by a preparative HPLC system (Agilent Technology, Santa Clara, CA, USA). The fractionated isoforms were completely dried in a SpeedVac vacuum drier (Eppendorf, Hamburg, Germany), and then dissolved in dimethyl sulfoxide (DMSO), which was effective regardless of pH (Abdel-Mawgoud et al., 2008). We used DMSO alone in a negative control experiment.

The HPLC-profile of surfactin produced in this study was quite clean and comparable to Sigma surfactin. The HPLC profile was provided as Supplementary information Control surfactin was also used in the cellulase activity assays as a positive control and included in Figure 4. 

Furthermore, fig. 4E states that Tris was used and not water (line 321 and figure legend).

our response) it was corrected.

It would be interesting to see the HPLC profile of surfactin isoforms. In line 32 it is said there are at least 4 isoforms; in line 184 it is stated that surfactin was fractionated into 4 isoforms and therefore it sounds odd to read “4 of the individual isoforms” in lines 33-34. Could you please show the data and, if it is the case, specify whether the 4 chosen are the “major isoforms”?

our response) Yes, they are the major isoforms a, b, c, d. The fractionation profile was included as supplemental data.

Also important is the recurrent quotation -and even a part of discussion – of a putatively accepted manuscript. Please remove the information that refers to unpublished data from the abstract (lines 21-23).

our response) We removed the relevant information in the abstract section. The same information was kept in the end of the Discussion section with the reference, Cho et al. (Cho et al., 2022)

Since it is stated that the manuscript has been accepted for publication (lines 370, 414), please quote it as a reference in the bibliography with the full list of authors and the journal (example at https://www.nlm.nih.gov/bsd/uniform_requirements.html#unpublished: “Tian D, Araki H, Stahl E, Bergelson J, Kreitman M. Signature of balancing selection in Arabidopsis. Proc Natl Acad Sci U S A. Forthcoming 2002.”). However, remove the paragraph from line 413 to 423 as it pertains to the discussion of other results that have nothing to do with the current manuscript.

our response) Citation was included and the last paragraph was removed. 

Minor comments:

Authors should describe how IC50 was calculated.

our response) The half-maximal inhibitor concentration (IC50)) was obtained using GraphPad Prism (GraphPad Software, 8.4.3, San Diego, CA, USA).

77-78: Could you clarify for readers (i.e., in the text) why “surfactant-secreting strains because would not trigger the emergence of resistant pathogens even after extensive use”? Which could be the other characteristics that can be screened that do trigger resistance, besides antibiotic production?

our response) Surfactin forms micelles in a hydrophilic environment and physically inserts in and disrupts the lipid-bilayer-cell-membrane, making it leaky. We speculated that it would be difficult to invent mechanisms to enforce the cell membrane to suppress surfactin insertion or sequester surfactin. Surfactin has been known for its antimicrobial, antifungal, antimicoplasmal, and antiviral effects. However, its development was hampered by its hemolytic effects at over a 100 mM concentration. The claims on the resistance pathogens were based on speculation, thus we changed the sentence to avoid unnecessary debates. (Line 79-80) we preferred to screen surfactant-secreting strains because surfactin also has anti-viral, anti-fungal, and anti-mycoplasma activities (Chen et al., 2015).

Reference 2: wrong journal name.

Corrected

56: gas emission. Could you please specify whether the reduction is supposedly derived from lower methane production from animals or less carbon dioxide from land use or reduced nitrous oxide from manure and slurry management? Please, quote a reference for this strong statement. our response) We elaborated on this sentence (Line 55-58): Important properties of probiotic strains for livestock are directly associated with enhancing productivity and animal welfare, and possibly the reduction of noxious gas emission derived from a decreased production in the gastrointestinal tract (Lan and Kim 2017).

65: “their effect” -> their probiotic effect

 ok

98: Bacillus subtilis should be in italics 

ok

115: cell (not plural)

ok

116: reference 9 is wrong, at least in this position

ok

122: substitute “Their” with “The “

ok

124: the accession number CP081458 does not provide any result in GenBank. Please check it.

Cp081458: Cp076445: https://www.ncbi.nlm.nih.gov/nuccore/CP076445

Cp076445.1: https://www.ncbi.nlm.nih.gov/nuccore/CP076445

124-25: Please provide the link to the “Korea Microbe Collection Center” and specific links to the accession numbers KACC81161BP 125 and KACC81159BP. It was impossible to find them through google search. 

http://genebank.rda.go.kr/, https://www.acm-mrc.asia/M/kctc.html

129-133: paragraph unclear for several reasons. Using the past tense and the singular verb (There were 10 operons in B. subtilis genome) I do not understand whether the information concerns one of the two strains newly described or 168 or else. The use of the past tense made me think that the description refers to the new strains, but then it should be plural… also, the rest of the paragraph is obscure. “All” means many. Which are the genomes that are under analyses here? Please, rephrase it.

our response) Changed (Line 134-136): The genomes of ps4100 and ps4060 each contained ten rRNA operons. The ten 16S rRNA sequences were mined from each genome and examined: only four of them had unique sequences.

135: add references to the list and substitute with numbers.

References were properly cited. (Miyazaki and Tomariguchi, 2019; van Berkum et al., 2003)

141: “two unknown strains”. Which is the source of these strains?

our response) They are not identified in this study, we decided to exclude them. 

150: representing?

Ok

161: incubated?

Ok

165: 5 µM refers to column diameter? Please specify

Ok, we elaborated.

177: significance of ΔΔ symbol should be given.

our response) The definition of ΔΔ was presented in the text (Line 193-194): We used ΔΔ to compensate for possible irregularity due to background absorbance in each well.

216: Which is the source of the antibodies for iNOS and actin?

our response) inducible Nitric Oxygen (iNOS; 1:1000 dilution; ABcam, Cambridge, UK, ab15323), β-actin (1:1000 dilution; Santa Cruz, CA, USA, sc-47778)

237: “We tested ~2,000 colonies for three months” means that the screening procedure took three months or it means that you repeated the same screening over and over for 90 days?

our response) We changed “for” to “over.”.

257-58: Could you show the “differences in the size of clear zones and growth patterns” for the two selected strains?

our response Line 282-285) The size of the clear zone, not the colony, was similar but became slightly larger around ps4060 than ps4100 when they grew over a four-day period at 30˚C. In contrast, the ps4060 colony was flat and about two times wider in diameter than the convoluted colony of ps4010.

264: than TO other

our response) We added “to” in front of the other Bacillus species.

268-270: “…each was respectively the closest relative to the B. subtilis SRCM103882 and BAB-1 groups (Fig 2B).” please rephrase.

our response) We elaborated as: Strain ps4100 was a close relative to B. subtilis SRCM103882 and ps4060 belonged to a group including the B. subtilis BAB-1 strain (Fig 2B). To further investigate their phylogenetic relationship, we built a phylogenetic tree with 780 orthologous genes in 12 selected taxa and performed bootstrap analyses (Fig 2C). The strains ps4100 and ps4060 were clearly separated into two distinct groups with strong bootstrap support (Fig 2D).

272: subtilis

Ok

278: please, enrich legend to Figure with more details. You should explain why each strain appears with different numbering (does it refers to different rRNA operons?). Define which method was used for the tree in A (maximum likelihood or others?)

our response Line 311-Line 328) 

Figure 2. Phylogenies. 

A. Maximum likelihood phylogeny constructed with 16S rRNA sequences. The numbers in parenthesis indicate the total number of unique sequences and total number of rRNA operons in each Bacillus species. In B. cereus, for example, eight of fourteen 16S rRNA sequences were unique. In each strain of ps4060 and ps4100, four of ten 16S rRNA sequences were unique. These sequences were identical between ps4060 and ps4100, so only ps4060 was marked on the tree. Numbers on branches represent branch length and bootstrap values. B. Maximum likelihood phylogeny constructed with 1,162 orthologous genes shared among all taxa. C. Maximum likelihood phylogeny of 770 orthologous genes among 13 selected taxa. Numbers below each branch indicate bootstrap values. D. Consensus tree after collapsing nodes with bootstrap support below 50%.

Figure 3. Global alignment of selected genomes.

Comparison of genome structures among eight strains closely related to ps4100 (A) and nine strains close to ps4060 (B). Identically colored blocks are locally collinear regions of multiple gene sequences without rearrangement of homologous genes. Each line connects a set of collinear blocks. Colored blocks below the centerline indicate inverted sequences.

312: legend to fig. 4E does not correspond to what shown in the figure.

Figure was corrected.

327 “thereby possibly enhancing” instead of which

Changed

329 please move “activated” after “cells”. Please account for the presence of LPS in fig. 5A & B and in the corresponding legend.

Moved

347: odd sentence “recommendations based on a guideline concerning…”.

our response Line 400-403) Thus, these strains are suitable probiotic candidates according to the regulation set by the Korean Food and Drug Administration, a guideline concerning the assessment of bacterial susceptibility to antimicrobials of human and veterinary importance (KFDA, 2021).

350: please change the title to Fig. 6 to a sentence referring to the scientific meaning of it (not to the method)

our response) Changed, Line 405-409: Figure 6. Susceptibility to six antibiotics. In each frame, the value on the left is the minimal inhibitory concentration (MIC) measured in this study; on the right is the acceptable MIC designated by the Korean Food and Drug Administration. The test was conducted using the Epsilometer test (E-test). Unit: ug/ml

357: natto

Ok

358: have been?

Removed “should”

366: prone to antibiotics?

Replaced with “susceptible”

367-368: “The possibility appeared to be very low to transfer antibiotic resistance from these strains to animals or farm workers”??? It was just stated that the strains do not contain resistance genes. Is this sentence meaningful?

our response) Line 422-424 was rephrased: They were novel strains (Fig 2 and 3), susceptible to antibiotics (Fig 6), thus eliminating concerns over transferring antibiotic resistance from these strains and suitable as probiotic candidates.

Reviewer #3: This manuscript describes how two new Bacillus subtilis strains were selected for screening probiotic candidates, taking into account surfactin secretion as a marker in vitro, and identified by genome sequencing. In addition, four surfactin isoforms are described, in which the cellulase activity and the mean maximal inhibitory concentration (IC50) were determined; as well as the susceptibility to several antibiotics of both strains. The article is also supported by another in vivo test in pigs (L21-23), which is reported to be in press, according to what is written by the authors in L370 and 414.

our response) A reference was cited once in the Discussion section but the description about the in vivo test was removed in other places. 

The topic is interesting, and the text is well structured and written. However, the authors can find here some suggestions and corrections. My main concern is about referencing properly to the publication related to the in vivo test. This study is under review or has already been accepted for publication. Where and who are the authors? Please, clarify and add this reference to the references section, detailing the year in the in-text citations.

Authors wrote (L103-104): “Halo quality was arbitrarily assessed by clarity, transparency, and evenness of oil layer in the clear zone.” Arbitrarily? Not very scientific, is it? How were clarity, transparency and uniformity evaluated? For example, using an increasing subjective 3- or 4-point scale.

our response) L105-109: Halo quality was visually assessed by clarity, transparency, and evenness of the oil layer in the clear zone, using three grades: 1) opaque with numerous small oil drops, 2) medium clear with a small number of oil drops, and 3) clear with no visible oil drops.

L120: “This process included consensus polishing with Quiver”. Please, add a citation.

our response) done (Chin et al., 2013)

L135: “(van Berkum et al., 2003; Yabuki et al., 2014)”. Please, pay attention since these citations are missing in the references section.

our response) References were properly cited (Miyazaki and Tomariguchi, 2019; van Berkum et al., 2003)

L166: “(Corning, Germany)” Which is the city? Please, be consistent with other resources cited.

(Corning, Berlin, Germany)

L197 “(Molecular Devices, Spectra Max 190)” Which is the location of company? See comment above.

microplate reader, Spectra Max 190 (Molecular Devices, San Jose, USA)

L220 “(Thermo Fisher, Rochford, USA)” Sometimes the U.S. city and state of the company are listed, but not this time. However, other times the country is not listed. Please, be consistent according to the journal guidelines. Check it throughout the M&M section.

Ok, 

L247: “Black circles indicate two colonies with larger, clearer halos than the control.” Black? They are red in the draft.

We changed this to “colored circle” instead of “black circle.”

L263: “Genome size and GC% were more similar to B. subtilis than other Bacillus species (Table 1)” Guanine-cytosine percentage? Acronyms should be described the first time they are used in the text (they should also be detailed in the tables and/or figures). Table 1, Table 2

Ok

Fig 2C and 2D are not mentioned in the text. Moreover, they both look identical. What are they used for?

Our response) We elaborated (Line 298-301): To further investigate their phylogenetic relationship, we built a phylogenetic tree with 780 orthologous genes in 12 selected taxa and performed bootstrap analyses (Fig 2C). The strains ps4100 and ps4060 were clearly separated into two distinct groups (Fig 2D).

L298: “Under the test conditions in this study, surfactin yields were respectively 0.3 and 0.2 mg/mL for strains ps4100 and ps4060, exceeding the 0.15 mg/mL yield of ATCC21322 mg/mL under the same conditions (Fig 4A)” Both strains cannot exceed 0.15 mg/mL because there is a difference of 0.1 mg/mL between these two lines. Rewrite this sentence accordingly. In addition, according to the figure 4A (with bars and lines), there appears to be some variation in the quantity produced. How was the difference between the amount of surfactin produced between two strains evaluated (statistical comparison to assess whether it was different from zero)? Figures should be self-explanatory. On the other, using the same range or amplitude of the y-axis of Figures 4B-E would improve the visual comparison.

Our response: Rewritten (Line 340-342): Under the test conditions in this study, surfactin yields were respectively 0.291 and 0.177 mg/mL for strains ps4100 and ps4060, which is more than 0.143 mg/mL yield of ATCC21322 under the same conditions (Fig 4A)

L350-353 and Figure 6. What were the units? A number is always followed by its unit.

Unit “ug/ml” was added.

L400: “The effects of surfactin on cellulases could be substantial if the quantity or Km is over ten-times higher than other enzymes.” Km??? Please, clarify.

Our response) Rephrased (Line 458-460): The effects of surfactin on cellulases could be substantial if the quantity of cellulase or affinity for its substrate is over ten-times higher than other enzymes.

References: Sometimes the name of the journal is abbreviated (e.g. Reference number 9, 17, etc.), sometimes it is not. Please, use journal names in ISO4 format. Check it throughout the section. In the reference 2, the name of journal is only Animals (remove : an open access journal from MDPI).

Abdel-Mawgoud, A.M., Aboulwafa, M.M., and Hassouna, N.A. (2008). Characterization of surfactin produced by Bacillus subtilis isolate BS5. Applied biochemistry and biotechnology 150, 289-303.

Bogdan, C. (2015). Nitric oxide synthase in innate and adaptive immunity: an update. Trends in immunology 36, 161-178.

Chen, W.-C., Juang, R.-S., and Wei, Y.-H. (2015). Applications of a lipopeptide biosurfactant, surfactin, produced by microorganisms. Biochem Engineering J 103, 158-169.

Chin, C.S., Alexander, D.H., Marks, P., Klammer, A.A., Drake, J., Heiner, C., Clum, A., Copeland, A., Huddleston, J., Eichler, E.E., et al. (2013). Nonhybrid, finished microbial genome assemblies from long-read SMRT sequencing data. Nature methods 10, 563-569.

Cho, S., Lee, D., Kang, B., Song, J.H., Lim, C.B., Yun, G., Cho, Y., and Kim, I.H. (2022). Novel Bacillus strains with high enzyme activity improve productivity of finishing pigs and reduce noxious gas emissions. Animal Nutrition under review.

Elshaghabee, F.M.F., Rokana, N., Gulhane, R.D., Sharma, C., and Panwar, H. (2017). Bacillus as potential probiotics: status, concerns, and future perspectives. Frontiers in microbiology 8.

Goldstein, F.W., Ly, A., and Kitzis, M.D. (2007). Comparison of Etest with agar dilution for testing the susceptibility of Pseudomonas aeruginosa and other multidrug-resistant bacteria to colistin. The Journal of antimicrobial chemotherapy 59, 1039-1040.

KFDA (2021). Guidelines for safety evaluation of probiotics, KFDA, ed. ( Korea Food and Drug Administration).

Leser, T.D., Knarreborg, A., and Worm, J. (2008). Germination and outgrowth of Bacillus subtilis and Bacillus licheniformis spores in the gastrointestinal tract of pigs. Journal of applied microbiology 104, 1025-1033.

Miyazaki, K., and Tomariguchi, N. (2019). Occurrence of randomly recombined functional 16S rRNA genes in Thermus thermophilus suggests genetic interoperability and promiscuity of bacterial 16S rRNAs. Scientific reports 9, 11233.

van Berkum, P., Terefework, Z., Paulin, L., Suomalainen, S., Lindstrom, K., and Eardly, B.D. (2003). Discordant phylogenies within the rrn loci of Rhizobia. Journal of bacteriology 185, 2988-2998.

Worthington, C.E. (1988). Worthington Enzyme Manual. 76-79 (Worthington Biochemical corporation).

---

## [Decision Letter · Decision Letter 1]

27 Oct 2022

Throughput screening of Bacillus subtilis strains that abundantly secrete surfactin in vitro identifies effective probiotic candidates

PONE-D-22-21217R1

Dear Dr. Yangrae Cho,

We’re pleased to inform you that your manuscript has been judged scientifically suitable for publication and will be formally accepted for publication once it meets all outstanding technical requirements.

Kind regards,

Joseph Selvin, Ph.D

Academic Editor

PLOS ONE

Additional Editor Comments (optional):

The Figure 2B is very complex, may be moved to supplementary doc. The legend require appropriate discerption for Western blot results

Reviewers' comments:

Reviewer's Responses to Questions

**Comments to the Author**

1. If the authors have adequately addressed your comments raised in a previous round of review and you feel that this manuscript is now acceptable for publication, you may indicate that here to bypass the “Comments to the Author” section, enter your conflict of interest statement in the “Confidential to Editor” section, and submit your "Accept" recommendation.

Reviewer #2: All comments have been addressed

2. Is the manuscript technically sound, and do the data support the conclusions?

Reviewer #2: Yes

3. Has the statistical analysis been performed appropriately and rigorously? 

Reviewer #2: Yes

4. Have the authors made all data underlying the findings in their manuscript fully available?

Reviewer #2: Yes

5. Is the manuscript presented in an intelligible fashion and written in standard English?

Reviewer #2: Yes

6. Review Comments to the Author

Reviewer #2: Please, correct LB= Luria BerTani (not Berani)

add Anti-iNOS before "primary antibody (1:1,000 dilution; ABcam, Cambridge, UK, ab15323)"

7. PLOS authors have the option to publish the peer review history of their article (what does this mean?). If published, this will include your full peer review and any attached files.

Reviewer #2: No
